# RNA-seq Based Transcriptome Analysis of the Anti-Obesity Effect of Green Tea Extract Using Zebrafish Obesity Models

**DOI:** 10.3390/molecules24183256

**Published:** 2019-09-06

**Authors:** Liqing Zang, Yasuhito Shimada, Hiroko Nakayama, Youngil Kim, Djong-Chi Chu, Lekh Raj Juneja, Junya Kuroyanagi, Norihiro Nishimura

**Affiliations:** 1Graduate School of Regional Innovation Studies, Mie University, Tsu, Mie 514-8507, Japan; 2Mie University Zebrafish Drug Screening Center, Tsu, Mie 514-8507, Japan; 3Department of Integrative Pharmacology, Mie University Graduate School of Medicine, Tsu, Mie 514-8507, Japan; 4Department of Bioinformatics, Mie University Life Science Research Centre, Tsu, Mie 514-8507, Japan; 5Rohto Pharmaceutical Co., Ltd., Osaka 544-8666, Japan; 6UMOU Science Lab, Mastusaka, Mie 515-0303, Japan

**Keywords:** green tea extract, visceral adipose tissue, obesity, zebrafish, RNA-seq

## Abstract

Green tea is a popular beverage that is rich in polyphenolic compounds such as catechins. Its major content, (-)-epigallocatechin-3-gallate, has been shown to have beneficial effects on several diseases including cancer, metabolic syndrome, cardiovascular diseases, and neurodegenerative diseases. The aim of this study was to assess the anti-obesity effects and the underlying molecular mechanisms of green tea extract (GTE) using zebrafish larva and adult obesity models. We administered 100 μg/mL GTE to zebrafish larvae and performed a short-term obesogenic test. GTE significantly decreased the visceral adipose tissue volume induced by a high-fat diet. Oral administration (250 µg/g body weight/day) of GTE to adult diet-induced obese zebrafish also significantly reduced their visceral adipose tissue volume, with a reduction of plasma triglyceride and total cholesterol levels. To investigate the molecular mechanism underlying the GTE effects, we conducted RNA sequencing using liver tissues of adult zebrafish and found that GTE may ameliorate the obese phenotypes via the activation of Wnt/β-catenin and adenosine monophosphate-activated protein kinase (AMPK) pathway signaling. In addition, the comparative transcriptome analysis revealed that zebrafish and mammals may share a common molecular response to GTE. Our findings suggest that daily consumption of green tea may be beneficial for the prevention and treatment of obesity.

## 1. Introduction

Green tea is one of the most famous and popular beverages globally and originated several thousand years ago in China, spreading first in Asia and then throughout the world. Followed by black tea, green tea is the world’s second-most consumed tea. Green tea is a non-fermented product that is rich in polyphenols, including catechins and flavonoids [1]. The major catechins in green tea are (-)-epigallocatechin-3-gallate (EGCG) (approximately 50–70% of the total catechins from green tea leaves), (-)-epigallocatechin (EGC), (-)-epicatechin-3-gallate (ECG), and (-)-epicatechin (EC) [2,3]. The health benefits of green tea catechins (especially EGCG) have been extensively studied, including anti-virus [4], antimicrobial [5], anti-cancer [6,7], anti-diabetes [8,9], anti-metabolic syndrome [10], and anti-cardiovascular disease effects [11]. Among the green tea catechins, EGCG has shown the highest biological activity and is known to have dual actions in relation to reactive oxygen species (ROS) as an anti-oxidant and a pro-oxidant [12,13]. In addition, *in vitro* and *in vivo* studies have revealed the anti-obesity effects of green tea [12,14]. Human trials also demonstrated that continuous ingestion of a green tea extract high in catechins significantly reduced body weight and body fat mass [15]. Human intervention and basic molecular studies have revealed that the anti-obesity function of green tea catechins was brought about by food intake reduction, regulation of lipid metabolism and increase in energy expenditure via thermogenesis, and fat oxidation and fecal lipid excretion [16].

The zebrafish is a well-characterized vertebrate animal model for developmental biology, drug discovery, toxicology, human genetics, and studies of human diseases [17,18,19]. It demonstrates a high similarity to the human genome and to human anatomy and physiology. Recently, studies on zebrafish have focused on its advantages related to metabolic syndrome research, including obesity and type 2 diabetes mellitus [20]. The optically semitransparent body provides a unique opportunity to observe whole organism adiposity in zebrafish larva by Nile red or Oil red O staining [21,22]. The similarity of lipid metabolism and adipogenesis in zebrafish to mammals has been demonstrated [23]. Adult zebrafish are also an excellent model for conducting studies on obesity [24]. Over-feeding-induced obesity in zebrafish shows similar symptoms to those in humans, such as hypertriglyceridemia and hepatosteatosis, and shares common pathophysiological pathways of obesity in mammals [24]. Overall, zebrafish constitute an attractive model to be used for evaluating the effects of functional foods and compounds on the development of obesity and its treatment [25,26,27].

In this study, we administrated green tea extract (GTE) for both larva and adult zebrafish obesity models to evaluate the anti-obesity effect of GTE. We further performed RNA sequencing (RNA-seq) of liver tissues from adult zebrafish to investigate the underlying molecular mechanism of GTE and to compare the gene expression profiles with those of mammals by studying mice as a model.

## 2. Results

### 2.1. GTE Decreased Visceral Adipose Tissue (VAT) Volume Induced by High-Fat Diet in Zebrafish Larvae

To confirm the anti-obese effects of GTE, we performed a zebrafish obesogenic test (ZO test) using zebrafish larvae, as previously reported [21]. The diagram of the experimental design is shown in Figure 1a. A Nile Red fluorescent probe was used to visualize and quantify lipid accumulation of visceral fat tissue in the larva. Nile Red staining was performed twice, before and after GTE administration (100 μg/mL), to evaluate the change of the VAT volume. After one day of high-fat diet (HFD) feeding, lipid accumulation was visualized in the posterior part of the intestine (Figure 1b). One day of starvation induced a decrease in VAT volume in the control group and a more remarkable reduction in the GTE-treated larvae (Figure 1c). The volume of VAT was quantified by Nile Red fluorescence signal, and a decrease in the VAT ratio was calculated (Figure 1d). GTE-treated larvae exhibited a significant decrease (−66%; *p* < 0.05) in HFD-induced VAT volume compared with control larvae.

### 2.2. GTE Reduced VAT and Decreased Plasma TG and TCHO Levels in Adult Obese Zebrafish

Next, we orally administered GTE (250 µg/g body weight/day) to adult zebrafish. The diagram of the experimental design is shown in Figure 2a. GTE was fed for 2 weeks, followed by overfeeding for one week. The short-term overfeeding for a week increased the body weights of the overfeeding (OF) zebrafish group compared to those of the normal-feeding (NF) group (*p* < 0.1; Figure 2b,c). Although there was no statistically significant difference in the change in body weight between overfeeding and GTE-treated groups, the GTE group showed reduced body weight gain (*p* = 0.26). Three-dimensional micro-computed tomography (micro-CT) analysis revealed that VAT volume in the overfeeding group was higher (*p* < 0.1) than that in the NF group; however, GTE significantly reduced the VAT volume (*p* < 0.05) compared with that in the overfeeding group (Figure 2d). To determine the effect of GTE on glycolipid metabolism in blood, we measured fasting blood glucose, plasma triglycerides (TG) and total cholesterol (TCHO) levels. GTE did not affect the fasting blood glucose levels (data not shown). The plasma TG and TCHO levels were significantly increased by overfeeding compared with those of the NF group (Figure 2e,f). In contrast, GTE administration resulted in a significant reduction in plasma TG and TCHO levels than those of the overfeeding group (*p* < 0.01 and *p* < 0.05, respectively). An independent replicate experiment also showed similar results (Appendix A).

### 2.3. GTE Exhibits Anti-Obesity Effects via Wnt/β-catenin and AMPK Pathways

To investigate the underlying molecular mechanism of the anti-obesity effects of GTE, we performed RNA-seq transcriptomic analysis of liver tissues from adult obese zebrafish with or without GTE-treatment. An enrichment false discovery rate (FDR) q-value <0.05 was considered statistically significant. According to the Ensemble gene ortholog database (http://www.ensembl.org/biomart/martview), we converted these altered genes to human orthologs and then conducted Gene Set Enrichment Analysis (GSEA) followed by Sub-Network Enrichment Analysis (SNEA) based on the algorithm of GSEA [28]. The sub-networks for protein expression targets revealed 12 gene sets (a priori defined sets of genes belonging to the same biological pathway) regulated by GTE (Appendix A). We also performed signal transduction pathway analysis [29]. In the Wnt canonical signaling pathway, Wnt/Frizzled (Wnt/Fzd) complex and its downstream genes, glycogen synthase kinase 3 beta (GSK3B) and β-catenin (CTNNB1), were upregulated by GTE (Figure 3a). The levels of gene expression of *gsk3b* and *ctnnb1* in the liver tissues of GTE-treated OF zebrafish were validated by qPCR analysis. GTE significantly upregulated the expression of *gsk3b* (*p* < 0.05) and showed a tendency (*p* < 0.1) to activate *ctnnb1* expression (Figure 3c,d). In addition, the adenosine monophosphate-activated protein kinase (AMPK) signaling pathway was also upregulated. Major downstream pathways of AMPK, such as the sterol regulatory element-binding transcription factor 1 (SREBF1, also known as SREBP1) pathway related to lipogenesis and the mechanistic target of rapamycin (mTOR) pathway, whose activation contributes to obesity, were downregulated by GTE (Figure 3b). Quantitative real-time PCR revealed that the mRNA expression level of *prkaa1* (ampk subunit 1) showed a trend towards an increase (*p* < 0.1), while *prkaa2* (ampk subunit 2) was significantly upregulated (*p* < 0.05) by GTE treatment (Figure 3e,f). On the contrary, the expression of *srebf1* was significantly downregulated by GTE (Figure 3g, *p* < 0.05). The ratios of gene expression of the WNT/β-catenin and AMPK signaling pathways obtained from RNA-seq analysis are shown in Appendix A.

### 2.4. Common Pathways of the Anti-Obesity Mechanism in GTE between Zebrafish and Mouse

To further evaluate whether the GTE-administered zebrafish share common transcriptome pathways with mammals, we downloaded the DNA microarray data of GSE77964 from the Gene Expression Omnibus (GEO) database as a reference gene expression profile. GSE77964 is a dataset containing 24 mouse samples (C57BL/6, male) after seven different treatments, which exhibits the expression patterns of non-alcoholic fatty liver disease genes and the efficacy of green tea on a hyperlipidemia mouse model. We selected the high-fat diet treated group (four samples) and the high-fat diet + 100 µg/g body weight/day GTE group (three samples), and then performed SNEA analysis. In the pathways of lipogenesis regulation in the liver, 15 out of a total of 23 genes showed similar alterations in the gene expression between zebrafish and mouse (Figure 4a). SNEA analysis for expression targets revealed that the gene sets for matrix metallopeptidase 2 (MMP2) and SRY-box 9 (SOX9) were altered by GTE treatment, and exhibited similar gene expression profiles between the two animal models (25 out of a total of 36 genes, and 18 out of a total of 22 genes are shown in the similar color, respectively) (Figure 4b,e and Figure 4c,f). The gene expression changes in zebrafish liver were validated by qPCR and the results showed that *mmp2* was significantly downregulated (*p* < 0.01), while *sox9* showed a trend towards decreased expression (*p* < 0.1) following GTE administration.

## 3. Discussion

We performed a short-term obesogenic test using larvae of zebrafish at a dose of 100 μg/mL GTE and found that the GTE exposure caused a rapid and significant decrease in adiposity compared with that of control zebrafish (Figure 1). No fatal or abnormal effects on the zebrafish larvae were observed (data not shown). Furthermore, oral administration of GTE to adult obese zebrafish significantly decreased VAT volume, plasma TG, and TCHO levels (Figure 2). A previous study has reported that 40 days of GTE exposure (0.005% solution) significantly suppressed the adiposity in adult zebrafish and altered the expression of lipid catabolism genes [14]. Although the administration method, dosage, and the experimental period were different between the two studies, consistent results were obtained, especially the suppressive effect of GTE on VAT. Our study signifies that pre-treatment by GTE before the induction of obesity could prevent its development, suggesting that daily consumption of green tea prevents an increase in the VAT and promotes health.

Genome-wide gene expression analysis using transcriptome sequencing allows a rapid expression of genes and quantifies both the known as well as novel transcripts. In our previous study, we used a combination of transcriptome and proteome analyses in our zebrafish model for obesity and successfully identified several novel pathways that are involved in human diseases [30]. Although the expression of an mRNA does not necessarily translate to a corresponding change in its protein level completely, it is still valuable for inferring the alteration following a chemical treatment. The Wnt/β-catenin pathway (also known as canonical Wnt signaling) is important for cell proliferation, differentiation, and tumorigenesis [31,32]. The central functional role of Wnt signaling is to regulate the phosphorylation and subsequent degradation of cytosolic β-catenin level, which determines the activation of Wnt responsive genes [33]. Recent studies suggest that Wnt/β-catenin signaling inhibits adipogenesis by blocking the induction of the lipogenesis-related genes, such as peroxisome proliferator-activated receptor-g (PPARG), and CCAAT/enhancer-binding protein-α (CEBPA) [34]. A previous report suggests that EGCG activates the WNT/β-catenin pathway, which leads to the inhibition of the adipogenic transcription factors and lipid-metabolizing enzymes, leading to intracellular lipid accumulation in 3T3-L1 cells [35]. Our in vivo experiment also similarly demonstrated that GTE treatment activated Wnt/β-catenin pathway signaling in zebrafish liver. We therefore concluded that the anti-adiposity effects of GTE may partially occur via the activation of the WNT/β-catenin pathway. Moreover, GTE upregulated the mRNA expression of AMPK and suppressed the downstream adipogenic genes such as SREBF1 (Figure 3b,e–g). This result is also consistent with a recent report in mice, which revealed that EGCG reduces obesity and white adipose tissue weight gain, partly through AMPK activation [36].

Comparative transcriptome analysis between our zebrafish and mouse study revealed that GTE downregulated the gene expression involved in de novo fatty acids synthesis and upregulated the genes involved in lipid oxidization in both the species (Figure 4). In addition, high similarities of MMP2 and SOX9-related pathways between zebrafish and mouse are observed. Among the two gene set seeds (key regulators of the observed differential response), the activity of MMP2 is reported to be inhibited by green tea catechins, which is consistent with our results [37]. MMP2 is a known gelatinase that is capable of degrading type IV collagen and is also one of the noninvasive biochemical markers of chronic liver diseases such as liver fibrosis [38,39]. MMP2 was demonstrated to be downregulated by GTE-treatment in the qPCR validation study (Figure 4g). Although we did not study the effects on liver fibrosis, we hypothesize that GTE might ameliorate the hepatic fibrosis induced by overfeeding via the downregulation of MMP2 expression. SOX9 is the primary chondrogenic marker that promotes chondrogenesis [40]. Stable SOX9 knockdown in undifferentiated rat mesenchymal stem cells resulted in a decrease in their proliferation rate and an increase in apoptotic activity [41]. SOX9 also acts as an adipogenesis inhibitor by repressing the expression of the adipogenic factors CEBPB and CEBPD [42]. A previous study showed that EGCG could effectively promote chondrocyte growth by upregulating the expression of SOX9 and other cartilage-specific gene expressions in rabbit articular chondrocytes [43]. In contrast, we found that GTE decreased the expression of SOX9, as revealed by both the RNA-seq and qPCR analysis (Figure 4f). We hypothesize that GTE may bind to or activate a receptor and thereby downregulate the downstream SOX9 gene. Further research is needed to prove this hypothesis. In summary, our data using the zebrafish model to study the effects of GTE show that zebrafish share several common molecular pathways with mammals, thereby demonstrating the utility of this animal model for similar biological studies.

In this study, we demonstrated that GTE significantly suppressed the accumulation of visceral fat in zebrafish larvae and ameliorated the obese phenotypes of adult zebrafish, including visceral adiposity and dyslipidemia. The pathway analysis of RNA-seq data revealed that GTE could suppress lipid accumulation through the activation of the WNT/β-catenin and AMPK signaling pathways. Comparative transcriptome analysis using liver tissues treated with GTE revealed that zebrafish and mammals (studied using a mouse model) share common molecular pathways.

## 4. Materials and Methods

### 4.1. Ethics Statement

All animal procedures were approved by the Ethics Committee of Mie University, were performed according to the Japanese animal welfare regulation act on ‘Welfare and Management of Animals’ (Ministry of Environment of Japan), and complied with international guidelines.

### 4.2. Zebrafish Strains and Maintenance

Zebrafish (AB strain; the Zebrafish International Research Center, Eugene, OR, USA) were raised and maintained under standard laboratory conditions at 28 °C with a light/dark cycle of 14/10 h [44]. Embryos were collected after fertilization and kept at 28 °C in 0.3X Danieau’s solution (17.4 mM NaCl, 0.21 mM KCl, 0.12 mM MgSO_4_, 0.18 mM Ca(NO_3_)_2_, and 1.5 mM 4-(2-hydroxyethyl)-1-piperazinyl-ethane-2-sulfonic acid (HEPES); pH 7.6). The fish were fed with GEMMA Micro 75–300 (Skretting, Fontaine- les-Vervins, France) based on their developmental stage and length.

### 4.3. Chemicals and Zebrafish Obesogenic Test in Zebrafish Larvae

Green tea extract (GTE; PF-TP80), with a total polyphenol content over 80% and catechin content over 70% (EGCG content exceeding 40%), was purchased from Pharma Foods International Co., Ltd. (Kyoto, Japan). For the experiments, GTE was dissolved in distilled water to make a 100 mg/mL stock solution. The working solution was obtained by preparing a 1/1000 diluted solution of the stock solution in 0.3X Danieau’s solution.

A three-day zebrafish obesogenic test was performed as previously described, with minor modifications [21]. The experimental design diagram is shown in Figure 1a. In brief, zebrafish larvae were fed a standard diet (Gemma Micro 75, followed by Gemma Micro 150) until they were used in the ZO test. Larvae with standard length (SL; the distance from the rostral tip to the base of the caudal fin) of approximately 10 mm were selected and were placed into a 6-well plate. Each well contained 5 mL 0.3X Danieau’s solution and five larvae. On the first day of the ZO test, larvae were fed with 0.1% hardboiled chicken egg yolk as a high-fat diet. On the second day, larvae were starved in the control group, and the GTE group was exposed to the GTE working solution. The volume of VAT was measured before and after starvation or GTE treatment by Nile Red staining.

For Nile Red staining, a Nile Red (Sigma-Aldrich, St. Louis, MO, USA) stock solution of 500 μg/mL was prepared in DMSO. Just before use, the working solution was obtained by 1/100 dilution of the stock solution in 0.3X Danieau’s solution. Larvae were exposed to the Nile Red working solution in the dark for 30 min at 28 °C. The fish were then rinsed twice with fish water for 5 min, anesthetized in 500 ppm of 2-phenoxyethanol (Wako Pure Chemicals, Osaka, Japan), and observed under a BZ-X710 fluorescence microscope (Keyence, Tokyo, Japan). The Nile Red-positive intensity was quantified using ImageJ software (National Institutes of Health, Bethesda, MD, USA).

### 4.4. Adult Zebrafish Treatment

For oral administration of GTE to adult zebrafish, a 10% GTE-containing zebrafish food was prepared as previously described [45]. Female adult zebrafish (3-months-old) were randomly assigned to three treatment groups with five fish per 2 L tank (n = 5 per group): 1) normal feeding group—the zebrafish were fed a normal diet (gluten granules) throughout the experimental period (3 weeks), and additionally fed 5 mg cysts/fish/day of *Artemia* (brine shrimp) from the second week; 2) over-feeding group—the zebrafish were fed a normal diet (gluten granules) for 3 weeks, and overfed with 60 mg cysts/fish/day of *Artemia* [24] from the second week; 3) GTE group—the zebrafish were fed GTE-containing food (250 µg/g body weight/day) for 3 weeks, and subsequently overfed from the second week. The feeding details are shown in Appendix A. The GTE-containing food was fed to zebrafish 30 min before *Artemia* feeding. During feeding, the tank water flow was stopped for 2 h. The leftover *Artemia* was removed once daily by vacuuming to avoid water pollution. The body length and body weight were measured every week. Blood glucose, plasma TG, and plasma TCHO levels were measured at the end of the experiment [46,47].

### 4.5. CT Measurement of VAT Volume

Zebrafish were euthanized by immersion in an ice-water bath (5 parts ice/1 part water at ≤4 °C) for ≥20 min, and a 3D micro-CT scan was performed using an in vivo System R mCT 3D micro-CT scanner (Rigaku, Tokyo, Japan) as described previously [14]. Three-dimensional images were reconstructed and viewed using i-View type R software (J. Morita Mfg, Kyoto, Japan), and visualized and analyzed using CT Atlas Metabolic Analysis ver. 2.03 software (Rigaku). Measurement of VAT volume was limited to the abdominal cavity, defined as the area between the cleithrum and the anus.

### 4.6. RNA Isolation, Library Construction, and High-Throughput Sequencing

Fish were subjected to laparotomy, immediately transferred into tubes containing 3 mL of RNAlater (Qiagen, Hilden, Germany), placed at 4 °C for 24 h, and then stored at −20 °C until use. The liver tissues of five fishes from each group were isolated by dissection using forceps, and the total RNA was isolated using TRIzol reagent (Life Technologies, Carlsbad, CA, USA). Total RNA content was measured using the Nano Drop 1000 (Thermo Fisher Scientific, Waltham, MA, USA), and the integrity of RNA samples was determined using an Agilent 2100 Bioanalyzer RNA 6000 Pico Chip (Agilent Technologies, Santa Clara, CA, USA). Ribosomal RNA was depleted using the RiboMinus™ Eukaryote System v2 kit (Thermo Fisher Scientific, Waltham, MA, USA) according to the manufacturer’s protocol. The integrity of the rRNA-depleted RNA was confirmed using the Agilent RNA Pico assay reagents and Bioanalyzer 2100 system (Agilent Technologies). The samples with a low RNA Integrity Number (RIN < 8) were removed from the next step according to the manufacturer’s protocol. Finally, each group consisted of four samples, and these were further used for RNA library construction that was performed at The Center of Genetics (Mie University, Japan) using the Ion Total RNA-seq Kit v2 (Life Technologies, Carlsbad, CA, USA) following the manufacturer’s instructions. Preparations containing bar-coded libraries were loaded into 318 chips and sequenced on the Ion PGM system (Life Technologies). Signal processing and base calling were performed with Torrent Suite software version 4.0.1. Adapter sequences were trimmed using the same software.

### 4.7. Bioinformatics Analysis of RNA-Seq data

Bioinformatics analysis was performed using CLC Genomics Workbench 11.0.1 (Qiagen, Germany). The raw reads were cleaned by trimming low quality sequences with quality scores of < 13. Data were trimmed and mapped to the zebrafish *Danio rerio* annotated genome, build GRCz10. The expression of each transcript was detected and normalized using the transcripts per million (TPM) algorithm. Expression data were exported from CLC Genomics Workbench in the form of Microsoft Excel Spreadsheets. After statistical tests, we performed Gene Set Enrichment Analysis (GSEA) and Sub-Network Enrichment Analysis (SNEA) using Pathway Studio 9.0 (Elsevier, Amsterdam, Holland) according to our previous study [48].

### 4.8. Real-time Quantitative Polymerase Chain Reaction (qPCR) Analysis

The synthesis of cDNA from 500 ng of total RNA was performed using a ReverTra Ace qPCR RT Kit (Toyobo, Osaka, Japan). Real-time qPCR was performed with the cDNA samples using the Power SYBR Green Master Mix (Applied Biosystems, Foster City, CA, USA), and the ABI StepOnePlus Real-Time PCR System (Applied Biosystems) according to the manufacturer’s instructions. The relative mRNA levels were determined using *bact* as an endogenous control gene. The sequences of the primers are shown in Appendix A.

### 4.9. Statistical Aalysis

Statistical analyses were performed using the Student’s t-test or one-way analysis of variance (ANOVA) with the Bonferroni–Dunn multiple comparison test, depending on the number of comparisons, using GraphPad Prism version 8 (GraphPad Software Inc., San Diego, CA, USA). A p-value less than 0.05 denoted a statistically significant difference.

## Figures and Tables

**Figure 1 molecules-24-03256-f001:**
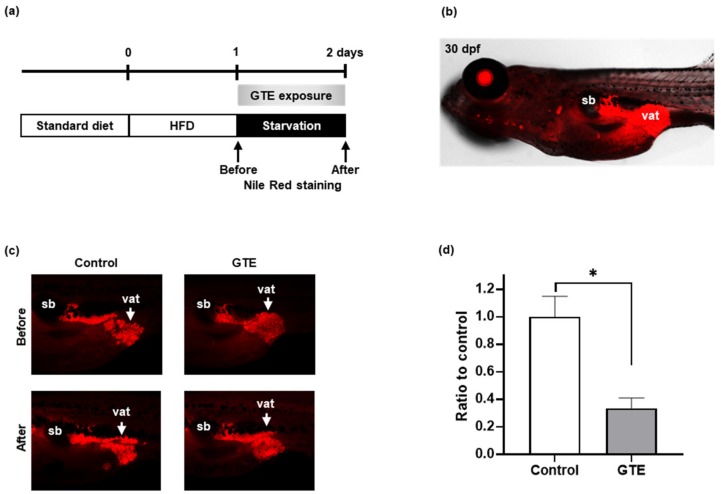
Green tea extract (GTE) suppressed lipid accumulation in zebrafish larvae. Live zebrafish at 30 dpf (day post fertilization) with a standard length of approximately 10 mm were used. (**a**) Experimental design of the zebrafish obesogenic (ZO) test. (**b**) Lateral view under a fluorescence microscope after Nile Red staining. vat, visceral adipose tissue (VAT); sb, swim bladder. (**c**) Representative VATs from live larva were selected after Nile Red staining, and images were recorded before and after 24 h of exposure or not to GTE (100 μg/mL). Nile Red-positive areas indicate VAT (red). (**d**) Quantification of the intensity of Nile Red positive staining. GTE reduced the amount of VAT compared to that in animals on a high fat diet (control). The Y axis indicates the ratio of Nile Red staining before and after GTE treatment for 24 h. * *p* < 0.05 vs. control, *n* = 5, error bars indicate SD.

**Figure 2 molecules-24-03256-f002:**
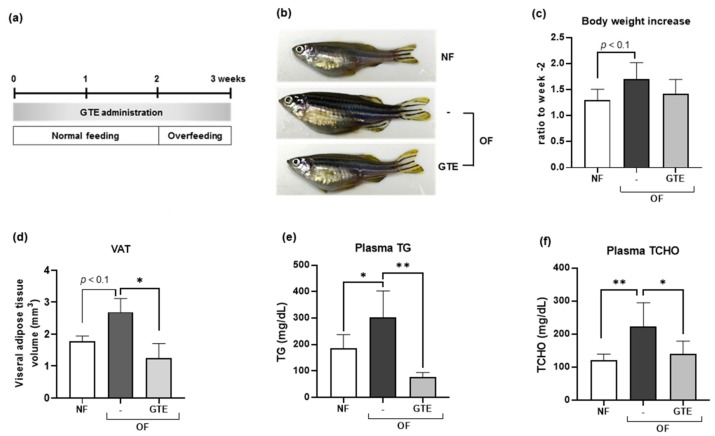
GTE reduced VAT and decreased plasma triglycerides (TG) and total cholesterol (TCHO) levels in adult obese zebrafish. (**a**) Diagram of the overfeeding experiment. GTE administration and overfeeding started on week 0 and week 2, respectively. (**b**) Typical images of normal-feeding (NF) and overfeeding (OF) zebrafish with or without GTE. (**c**) Change in body weight during the feeding experiment. (**d**) GTE reduced VAT in OF zebrafish. GTE suppressed the increase of plasma TG (**e**) and TCHO (**f**) in OF zebrafish. * *p* < 0.05, ** *p* < 0.01, *n* = 5, error bars indicate SD.

**Figure 3 molecules-24-03256-f003:**
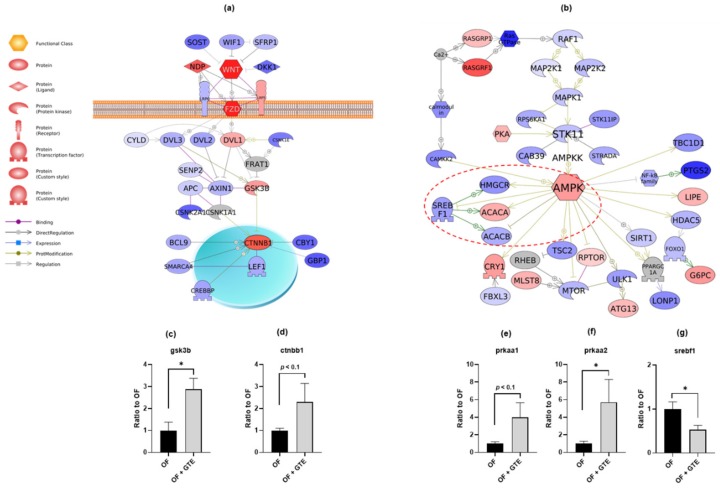
GTE ameliorates obesity in adult zebrafish by activating the WNT/β-catenin and AMPK signaling pathway. Biological pathways of (**a**) Wnt canonical signaling and (**b**) AMPK signaling were identified by Pathway Studio using altered genes in GTE-treated obese zebrafish compared with those of OF zebrafish. The red and blue colors denote genes with increased and decreased expression, respectively. Grey denotes the genes that were not detected in the RNA-seq assay, including FRAT regulator of WNT signaling pathway 1 (FRAT1), casein kinase 1 alpha 1 (CSNK1A1), Ras homolog, mTORC1 binding (RHEB) and PPARG coactivator 1 alpha (PPARGC1A). Gene expression changes of *gsk3b* (**c**), *ctnbb1* (**d**), *prkaa1* (**e**), *prkaa2* (**f**), and *srebf1* (**g**) were validated by qPCR analysis. The red dotted line indicates the sterol regulatory element-binding transcription factor 1 (SREBF1) branch of the AMPK pathway. * *p* < 0.05 vs. OF group, *n* = 4, error bars indicate SD.

**Figure 4 molecules-24-03256-f004:**
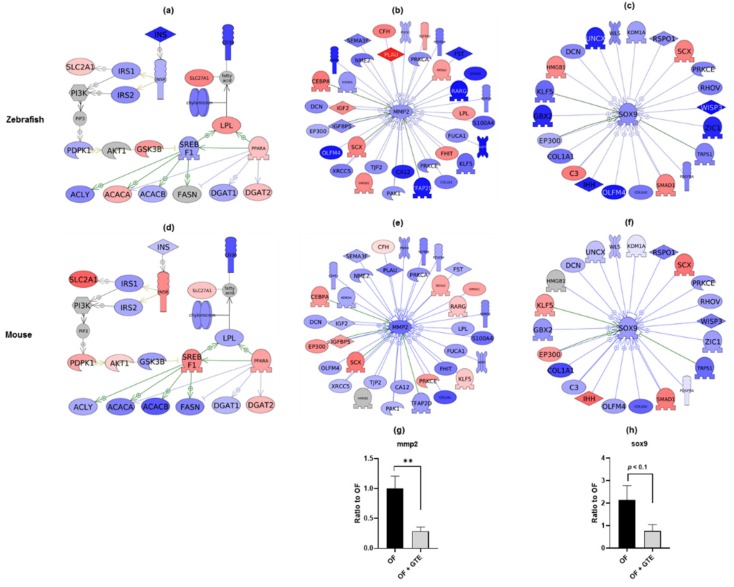
Gene expression profiles of liver tissues from zebrafish and mice treated with GTE. Pathways of lipogenesis regulation in adipocytes (**a**,**d**), protein regulators of MMP2 (**b**,**e**), and protein regulators of SOX9 (**c**,**f**) were identified by Pathway Studio using altered genes in zebrafish and mouse (GSE77964). Upper panels show the identified pathways in the zebrafish, and lower panels show pathways in mouse. The red and blue colors denote genes with increased and decreased expression, respectively. Grey denotes genes that were not detected in the RNA-seq assay. Gene expression changes of *mmp2* (**g**), and *sox9* (**h**) were validated by qPCR analysis. ** *p* < 0.01 vs. OF group, *n* = 4, error bars indicate SD.

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
