# Peer review of "RNA-seq Based Transcriptome Analysis of the Anti-Obesity Effect of Green Tea Extract Using Zebrafish Obesity Models"

_molecules, 2019, doi:10.3390/molecules24183256_

Round 1

Reviewer 1 Report

Manuscript ID: molecules-538032Title: RNA-seq based transcriptome analysis of the anti-obesity effect of green tea extract using zebrafish obesity models.

This manuscript is to provide information on the anti-obesity effect of the green tea extract (GTE) using zebrafish and the transcriptome analysis. The results are very interesting and would contribute greatly to the future study of the mechanism of beneficial health effects of green tea. However, there are deficits and before publication additional experiments would be required. 

1) The dose-dependent anti-obesity effects were not examined in the larva and adult experiments. It is not clear why 100 μg/mL and 250 μg/g BW/day were selected. Are the data derived from the three sets of  experiments? 

2) The amount of ingested GTE in the adult zebrafish may not be accurate because of “leftover”.

3) Is there any reason why the authors selected the different number of the sample for the high-fat diet (4 samples) and GTE groups (3 samples)

4) A sentence beginning from “Overfeeding-induced obesity” needs citation of Reference.

5) Abbreviations should be used according to the Journal’s instructions. For example, EGCG in the Abstract is used only once. It is not clear whether “RNA-seq” is Abbreviation.

6) In Figure 3, Grey color is not clear. Please describe the genes which were not detected in the RNA-Seq assay. 

7) There are some grammar errors. For example, in Abstract Our finding suggest ---.

Since this reviewer wants to have data based on the additional experiments, the manuscript at this stage is considered to be rejected. 

Reviewer 2 Report

See the attached document.

Reviewer 3 Report

The authors describe a nice little study where they investigate the effect of green tea extract on obesity in a zebrafish model. Using larvae, they present a quick experimental design that may have future applications in drug screens. Using adults, they identified a potential molecular mechanism underlying the anti-obesity effect of GTE, and elucidate one or two key proteins in this (MMP2 and SOX9). 

Although the study has merit based on scientific interest, there is a (big) flaw in the experimental design (1), and additional replicates are needed (2):  This makes me wonder if the study has potential for publication all together. 

1)    There is a flaw in the experimental design, as overfed control group is overfed with artemia only, and the experimental group was given GTE-loaded gluten granules, followed by artemia. Ideally, the authors would have given the overfed control group unloaded gluten granules, to control for differences in nutrient composition. 

2)    Feeding experiments with zebrafish are also very dependent on social behavior. Social behavior can differ remarkable between different groups of 5 fish. We have seen this often in our studies. The authors are encouraged to reproduce their results in at least 2 more groups of 5 fish, using e.g. qPCR for the most prominent genes identified in RNAseq. In feeding and behavioral studies, a single tank of multiple fish is usually seen as a single replicate. 

Other major comments:

RNA seq data appears to be over interpreted. The paper would be much stronger if specific readouts of pathways identified in the RNAseq dataset were validated with targeted studies (e.g. westernblot, qPCR validation of relevant WNT and AMPKK regulated genes)

Figure 3: 

1)    authors give a complete signaling pathway, in which they highlight changes in gene expression. However, to what extent is the signaling pathway regulated at the level of gene expression? Expression levels do not always correlate with protein levels, and things like phosphorylation add another layer of regulation.  I think that the authors overinterpret these data, as they give designations like “protein”, “protein kinase”, “modification”, and “binding” in the graphical overviews. It is not clear of protein levels decrease, and if the binding or modification is really altered in the tested samples. 

2)    Validation of RNAseq data with qPCR is lacking. Or were all livers analysed with RNAseq? This is not clear in the methods section. 

Discussion: how does the 100ug/ml GTE dose administered to larvae relate to human green tea intake? This is important to know to determine the value of this study, and whether the data support the conclusion that daily consumption of green tea can prevent an increase in VAT. 

Discussion: what is the effect of the fasting on the larval experiment. It is a surprising experimental design for me. Thinking of disease models, people are not likely to fast for a whole day. Also, fish survive fasting a lot better (and longer) then humans. 

Discussion: there is no real discussion on what the role of MMP2 could be in this! It presents as an interesting target, but it is best known as a gelatinase, and I do not see how it fits in this work. Some more insights should be provided.

Minor:

Line 28: “therapeutic mechanisms” it is not a therapy yet, so I think it is better to write about the underlying molecular mechanism.

Line 32: mammals (plural) AND “pathophysiological pathways when treated with GTE”. GTE does not induce a pathology, so mouse and zebrafish may share a common molecular response (or something along those lines, but please do not use “pathophysiological”)

Line195: SOX9 is not a target for catechins, SOX9 is a downstream regulated gene. Likely, catechins target (bind/activate) another molecule, such as a receptor. Please use better terminology. 

Figure 1a: please give the age/size of selected larvae at the start of the experiment in the figure. 

Figure 1C: catechin: authors use GTE in all others, so please be consistent. 

Section 4.2: what is the normal diet that zebrafish are given? 

Section 4.6: how many livers were sent for RNA seq per treatment?

Round 2

Reviewer 1 Report

Manuscript ID: molecules-538032; 

Most of the previous comments of this reviewer have been responded properly so that the manuscript can be accepted. However, The authors should have been more careful about the comments on Abbreviation: VAT, TG, TCHO, RNA-Seq in Abstract should be removed. “RNA-Seq” is defined as transcriptome analysis in line 66, but “RNA-Seq” is doubly defined as transcriptome sequencing in lines 117 and 172. “transcriptome analysis” are used without being abbreviated in lines 207 and 234. “micro-CT” is not defined in line 98. Thus, these examples make the content unclear so that authors should scrutinize carefully the whole text.

Author Response

Thank you for your valuable suggestions.

We have deleted the abbreviations in Abstract section, including VAT, TG, TCHO, RNA-seq.  

“RNA-Seq” should be defined as “RNA sequencing”. We have revised the abbreviation in line 66 and deleted the definition in lines 117 and 185.

Since “transcriptome analysis” has no abbreviation, we left it unchanged.

We have revised “micro-CT” as “micro-computed tomography (micro-CT)”.  

We read through the whole manuscript and revised several abbreviations such as OF and ZO test.

Reviewer 3 Report

Dear authors,

I am pleased with the amount of additional data and details the authors provided. Every feeding study in fish suffers from a few difficult-to-control parameters, such as the amount of food eating by each fish.
While you can not easily overcome these limitations, you have made significant improvements to the experimental design and description of the design.  I think that it can be published in its current form.

Best wishes

E

Author Response

Dear Reviewer,

Thank you very much for peer-reviewing our manuscript and valuing our work. Your valuable suggestions elevated the quality of our manuscript. We really appreciate your help.

Sincerely yours,

Liqing